# CONTENT-RICH AIGC VIDEO QUALITY ASSESSMENT VIA INTRICATE TEXT ALIGNMENT AND MOTION-AWARE CONSISTENCY

## ABSTRACT

The advent of next-generation video generation models like *Sora* poses challenges for AI-generated content (AIGC) video quality assessment (VQA). These models substantially mitigate flickering artifacts prevalent in prior models, enable longer and complex text prompts, and generate longer videos with intricate, diverse motion patterns. Conventional VQA methods designed for simple text and basic motion patterns struggle to evaluate these content-rich videos. To this end, we propose **CRAVE** (Content-Rich AIGC Video Evaluator), specifically for the evaluation of Sora-era AIGC videos. CRAVE proposes the multi-granularity text-temporal fusion that aligns long-form complex textual semantics with video dynamics. Additionally, CRAVE leverages the hybrid motion-fidelity modeling to assess temporal artifacts. Furthermore, given the straightforward prompts and content in current AIGC VQA datasets, we introduce **CRAVE-DB**, an *ITU-Compliant* benchmark featuring content-rich videos from next-generation models paired with elaborate prompts. Extensive experiments have shown that the proposed CRAVE achieves excellent results on multiple AIGC VQA benchmarks, demonstrating a high degree of alignment with human perception. All data and code will be publicly available to foster future research.

## 1    INTRODUCTION

Recently, text-driven video generation Brooks et al. (2024); Hunyuan (2024) has seen significant growth. However, evaluating these text-driven AI-generated videos presents unique and escalating challenges. These challenges primarily stem from two key issues: (1) the need for precise video-text alignment, especially with complex and lengthy text prompts; (2) the occurrence of distinct distortions that are not typically found in natural videos, such as irregular motion patterns and objects.

With the advancement of new-generation models, these challenges have become even more pronounced. These new-generation models, marked by the advent of Sora Brooks et al. (2024), offer substantial improvement in visual quality, characterized by rich details and content, such as Kling Kuaishou (2024), Gen-3-alpha Runway (2024), Vidu Shengshu (2024), etc. Compared with prior AIGC videos, they support **much longer and more intricate text (often over 200 characters), along with more complex motion patterns with longer duration (often over 5 seconds with the fps of 24)**. As illustrated in Figure 1, these rich contents impose greater demands on the evaluator's ability to understand video dynamics and the alignment with complex textual semantics.

To address this, we introduce Content-Rich AIGC Video Evaluator (CRAVE) to assess the quality of these next-generation text-driven videos. CRAVE evaluates videos from three perspectives: It firstly considers the traditional visual harmony, like the previous Video Quality Assessment (VQA) method Wu et al. (2023a), which measures the aesthetics and distortions. Furthermore, CRAVE leverages a multi-granularity text-temporal fusion module to align the intricate texts with video dynamics. Additionally, CRAVE incorporates the hybrid motion-fidelity modeling that exploits hierarchical motion information to assess the temporal quality of the next-generation AIGC videos.

Besides, the gap between the naturalness and complexity of the latest AIGC videos and previous videos has become markedly apparent. To better assess current AIGC videos, we introduce CRAVE-DB, a content-rich AIGC VQA benchmark consisting of elaborate text-driven videos generated by

**A polar bear** playing **drum kit** in NYC Times Square, **4k**, **high resolution**.

A **stylish woman** walks down a **Tokyo street** filled with **warm glowing neon** and **animated city signage**. She wears **a black leather jacket**, **a long red dress**, and **black boots**, and carries **a black purse**. She wears **sunglasses** and **red lipstick**. She walks **confidently** and **casually**. the street is **damp** and **reflective**, creating a mirror effect of the **colorful** lights. Many **pedestrians** walk about.

Figure 1: Comparison of concurrent and previous AIGC videos. Videos are generated by Lavie Wang et al. (2023d) (1st row) and Sora Brooks et al. (2024) (2nd row), respectively. Nouns that should be present in the video are highlighted in orange, while adjectives with more details are highlighted in blue. The new-generation AIGC videos contain richer content.

advanced models such as Kling Kuaishou (2024), Qingying Zhipu (2024), Vidu Shengshu (2024), and Sora Brooks et al. (2024). Here, by "elaborate text," we mean prompts that include complete descriptions of the subject, actions, and environment, with at least **5** detailed descriptions for any one aspect and a total character count exceeding **200**. These videos have largely eliminated issues prevalent in previous generations, such as flickering, weak motion, and short content. They encompass diverse scenes, subjects, actions, and rich details, with a duration of over 5 seconds and a frame rate of 24 fps. Extensive experiments show that CRAVE has achieved leading human-aligned video quality assessment results across multiple metrics on T2V-DB Kou et al. (2024b), currently the largest AIGC VQA dataset, and the proposed CRAVE-DB.

To summarize, our main contributions are as follows: (1) We introduce CRAVE, the effective evaluator for content-rich videos derived from the new-generation video models, which assesses AIGC videos from the temporal and video-text consistency via effective motion-aware video dynamics understanding and a multi-granularity text-temporal fusion module. (2) Given the gap between new-generation AIGC videos and previous ones, we introduce CRAVE-DB, a benchmark containing AIGC VQA samples produced by advanced models like Kling, etc., to facilitate the evaluation of contemporary content-rich AIGC videos. (3) Extensive experiments demonstrate the proposed CRAVE achieves excellent results on multiple AIGC VQA benchmarks with varying sources of videos and prompt lengths, showcasing a strong understanding of the quality of AIGC videos.

## 2 RELATED WORK

### 2.1 MEASUREMENT FOR TEXT-TO-VIDEO MODELS

Currently, common methods of evaluating text-driven generated videos include some objective metrics Radford et al. (2021); Unterthiner et al. (2018); Salimans et al. (2016) and human-aligned methods Kirstain et al. (2023); Qu et al. (2024); Kou et al. (2024b). Objective metrics such as CLIP-score Radford et al. (2021) measure the mean cosine similarity between the text and each frame. IS Salimans et al. (2016) utilizes the inception feature to measure the overall quality of image and video frames. However, these objective metrics do not align with human subjective perception and often evaluate videos from a single dimension. Some measurements for natural videos provide human-aligned overall evaluations Wu et al. (2023a; 2022); Kou et al. (2023). DOVER Wu et al. (2023a) assesses quality in terms of aesthetics and technicality. FastVQA Wu et al. (2022) utilizes grid mini-patch sampling to assess videos efficiently while maintaining accuracy. Q-Align Wu et al. (2023b) transforms the VQA task into the generation of discrete quality level words via the Multimodal Large Language Model. StableVQA Chai et al. (2023) measures video stability by separately obtaining the raw optical flow, semantic, and blur features. These are suitable for natural video quality assessment but do not consider the text-video alignment, which is key to the evaluation of text-driven videos. To address this, EvalCrafter Liu et al. (2024) establishes a series of indicators including CLIP score, SD score, and natural video assessment methods. T2V-QA Kou et al. (2024b) incorporates a transformer-based encoder and a Large Language Model to assess text-driven AIGC

videos. TriVQA Qu et al. (2024) explores the video-text consistency through cross-attention pooling and the recaption of Video-LLaVA. However, there are still relatively few VQA methods specifically for AIGC videos. With the growth of new-generation videos, the requirements for understanding video dynamics and text consistency are becoming increasingly demanding, posing greater challenges.

## 2.2 TEXT-TO-VIDEO GENERATION METHOD

Recently, lots of video generation models based on Rombach et al. (2022); Ho et al. (2020) have emerged Singer et al. (2023); Wang et al. (2023c;a); Blattmann et al. (2023); Chen et al. (2023a); Zheng et al. (2024); Lab & etc. (2024). They represent a significant breakthrough in video generation. However, videos produced by prior methods still tend to suffer from issues such as low resolution, short duration, flickering, and distortion. With the advent of Sora Brooks et al. (2024), the new-generation models Hunyuan (2024); LumaLabs (2024); MiniMax (2024); Tongyi (2024); Labs (2024); Yang et al. (2024) have made notable progress. Particularly recently, methods like Kling Kuaishou (2024), Gen-3-alpha Runway (2024), and Qingying Zhipu (2024) have achieved impressive video generation results and have been made available for community testing. These videos generally alleviate the foundational problems seen in previous methods, with a duration of more than 5 seconds and frame rates above 24 fps. Meanwhile, the content in these videos includes a lot of details, and they support the control via longer text inputs. Under the wave of new-generation video generation models, effectively assessing more complex spatiotemporal relationships within the videos and exploring their consistency with longer texts is a topic worthy of further study.

## 2.3 TEXT-TO-VIDEO VQA DATASET

Currently, there are still relatively few text-to-video QA datasets suitable for evaluating current AIGC videos. Eval-Crafter Liu et al. (2024) collects 700 prompts and uses 5 models to generate 2500 videos in total. FETV Liu et al. (2023) utilizes 619 prompts to generate 2,476 videos by 4 T2V models. Chivileva Chivileva et al. (2023)

Table 1: Comparison of prompt density and annotations.

| Dataset | # Words | # Chars. | # Ann. |
|---|---|---|---|
| FETV Liu et al. (2023) | 10.94 | 59.60 | 3 |
| EvalCrafter Liu et al. (2024) | 12.33 | 69.77 | - |
| T2V-CompBench Sun et al. (2024a) | 10.42 | 56.42 | 3 |
| VBench Huang et al. (2024b) | 7.64 | 41.95 | - |
| VideoGenEval Zeng et al. (2024) | 33.00 | 202.02 | 0 |
| T2VQA-DB Kou et al. (2024b) | 12.32 | 76.22 | 27 |
| **CRAVE-DB (Ours)** | **68.38** | **411.30** | **29** |

derives 1,005 videos generated from 5 T2V models. VBench Huang et al. (2024a) uses nearly 1,700 prompts and 4 T2V models to generate 6984 videos. T2VQA-DB Kou et al. (2024a) contains 10,000 videos generated by 1000 prompts. These datasets mainly meet two challenges: (1) According to the ITU-standard Series (2012), the number of human annotators should exceed 15 to keep the assessment error within a controllable range. Among these, only T2VQA-DB Kou et al. (2024a) and Chivileva Chivileva et al. (2023) meet the standard with 27 and 24 annotators. (2) The gap between the prior and the concurrent AIGC videos. Prior videos often involve only easy movements and commonly have basic issues such as flickering, which are relatively rarely seen in the new-generation video models. In this work, to address the issue that prior VQA datasets do not cover concurrent AIGC videos, we introduce CRAVE-DB, which focuses on next-generation AIGC videos with subjective scores from **29** annotators, to provide a robust assessment of concurrent AIGC videos.

## 3 CONTENT-RICH AIGC VQA BENCHMARK

With the advancements in text-driven video generation, there exists a significant gap between the concurrent and previous models in terms of visual quality, content complexity, and cross-modal understanding, as shown in Figure 1. These models have substantially alleviated basic issues such as flickering prevalent in earlier models, and have removed the prior length limitation of 77 tokens for input text. The challenges now shift towards content distortion in more complex spatiotemporal scenarios and semantic alignment with more intricate texts. However, current AIGC VQA datasets are still based on the previous generation of general models, creating a significant gap compared to the concurrent content-rich models. To this end, we introduce CRAVE-DB, a new AIGC VQA benchmark featuring intricate text prompts, content-rich videos generated by SOTA generation models, and the corresponding human scores. CRAVE-DB incorporates 410 intricate prompts, each containing dense text information, as shown in Table 1. Each video has a duration of over 5 seconds with a fps of 24.

In the subjective study, to ensure the data quality, we engage 29 humans to rate each video, nearly twice the minimum number of subjects required by the ITU standard Series (2012), to minimize the impact of variance in the ratings. From these, we screen out 35,612 Ratings ratings with 1,228 high-quality annotated videos to further enhance the quality of the dataset. We will subsequently introduce the prompt collection, video generation, and subjective study in detail.

## 3.1 PROMPT COLLECTION

In the past AIGC VQA datasets composed of prior-generation models, most supported prompt length is limited by CLIP Radford et al. (2021). In this case, these prompts tend to be brief, making it challenging to incorporate complex descriptions and scene compositions. For instance, we present the prompt density (average word and character count per prompt) of different datasets, as shown in Table 1. We could learn that most prompts in previous datasets contain merely a dozen words. This inherent limitation poses significant challenges for models when evaluating more sophisticated semantic alignment.

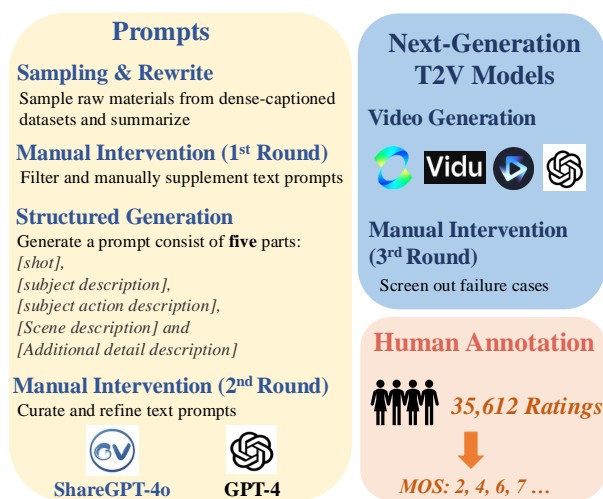

Figure 2: The collection pipeline of CRAVE-DB.

To address this, we propose the prompt generation pipeline in Figure 2. To ensure the prompts are detailed and semantically rich, we focus on the dense-captioned dataset, ShareGPT-4o Chen et al. (2023b), which leverages the advanced multimodal capabilities of GPT-4o to describe videos in detail. It contains rich annotations that even require summarization to be clear prompts. We randomly sampled 300 captions and summarized them using GPT-4 Achiam et al. (2023), retaining only key details. We then conducted the 1st intervention to filter out failed, redundant, or illogical generations.

Given that ShareGPT-4o primarily focuses on daily life scenarios, we manually crafted 200 more prompts to broaden the coverage of actions, subjects, and scenes. Prompts contain 4 categories: landscape, object, animal, and human. The "landscape" contains common scenes (e.g., grasslands, streets), rare environments (e.g., volcanoes, auroras), and renowned landmarks. The "animal" includes various mammals, reptiles, birds, fish, and amphibians. The "object" covers common real-world items, while the "human" features people across ages, genders, occupations, and clothing.

Subsequently, we employed GPT-4 to structure raw prompts using a template format: "[shot language] + [subject description] + [subject action description] + [scene description] + [additional detail description]". The "shot language" incorporates various cinematographic techniques including tilt shots, flat shots, progressive shots, surround shots, close-ups, and panoramic views. The scene descriptions encompass natural landscapes under diverse weather and lighting conditions. Following this, we initiated a second round of manual intervention to screen and refine all prompts, ultimately finalizing a curated set of 410 high-quality prompts. The overall word cloud is shown in Figure 3.

## 3.2 VIDEO GENERATION

Since the advent of Sora Brooks et al. (2024), text-driven video generation methods have achieved significant advancements in visual quality, text understanding, and the diversity and complexity of generated content. Given the substantial gap between current AIGC videos and prior ones, constructing datasets using the next-generation video models is essential. In this work, we employ Sora and other subsequent state-of-the-art models: Kling Kuaishou (2024), Vidu Shengshu (2024), Qingying Zhipu (2024) to build samples. As Sora had not been publicly released by the time of our subjective evaluation, we curated 14 content-rich prompts and their corresponding outputs from Sora's publicly showcased videos. All videos exceed 5 seconds in duration with a frame rate of 24 fps, and resolutions ranging from $384 \times 688$ to $960 \times 1440$, depending on the generation model. Considering the limited availability of publicly accessible AIGC VQA datasets that meet ITU

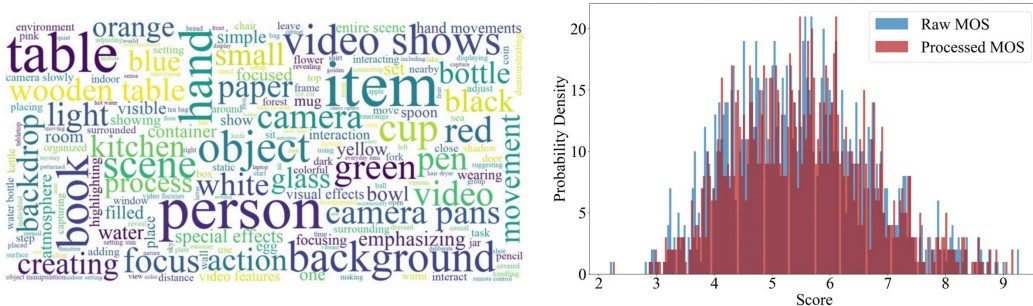

Figure 3: Word Cloud (left) and MOS distribution (right) in CRAVE-DB.

standards (with at least 15 annotators per sample), to evaluate newer video generation models and validate CRAVE's 0-shot generalization capability, we further constructed a test set based on 465 samples from VideoGenEval Zeng et al. (2024). VideoGenEval continuously keeps track of the latest video generation models, including Wan Wang et al. (2025), Hunyuan Hunyuan (2024), etc., and has relatively complex prompts (as shown in Table 1), which are suitable for 0-shot testing.

## 3.3 SUBJECTIVE STUDY

According to ITU Series (2012) standards, subjective experiments should involve at least 15 participants to reduce error fluctuations. To obtain the Mean Opinion Score (MOS) for each video, in our experiment, each video was scored by 29 different human subjects. They come from diverse backgrounds, including science, engineering, business, law, etc., with all being over 18 years old. Before scoring, all people were gathered on-site for training. During the training, we presented some cases out of the dataset, including good, bad, and average examples, to ensure a basic understanding of the task. There is a mandatory 5-minute break after every 15 minutes of scoring to prevent fatigue. Based on prior work Sun et al. (2024b), the scoring system used a 10-point scale. More details are included in Appendix H. **To ensure a fair comparison with the baseline** Kou et al. (2024b) **that rely solely on an overall score, and to implicitly model the relationship between dimensional scores and the final overall score**, the subjects were asked to provide a final overall score after considering all three aspects. After all the scoring was completed, we obtained a total of 35,612 subjective scores, from which we derived the raw MOS score. As shown in Figure 3, we then normalize the raw scores as Z-score MOS, which could be formulated as:

$$MOS_z(m,i) = \frac{X_{m,i} - \mu(X_i)}{\sigma(X_i)},$$

(1)

where $X_{m,i}$ and $MOS_z(m,i)$ denotes the raw and Z-score MOS of $m$-th video of $i$-th annotator, respectively. $\mu(\cdot)$ and $\sigma(\cdot)$ refer to the mean and standard deviation function, respectively. $X_i$ represents all MOS from the $i$-th annotator. After deriving $MOS_z(m,i)$, the screening method in BT.500 Int.Telecommun.Union (2000) is deployed to filter the outliers.

## 4 CONTENT-RICH AIGC VIDEO EVALUATOR

### 4.1 OVERALL FRAMEWORK

CRAVE evaluates videos from 3 perspectives: (1) visual harmony, measured using traditional metrics like aesthetics and distortion, (2) text-video semantic alignment, achieved via Multi-granularity Text-Temporal (MTT) fusion, and (3) motion-aware consistency, which is specific for dynamic distortions in AIGC videos, captured through Hybrid Motion-fidelity Modeling (HMM). The overall framework is illustrated in Figure 4. We will delve into the details of each module in the following sections.

### 4.2 VISUAL HARMONY

For traditional natural video quality assessment, we utilize DOVER Wu et al. (2023a) to assess individual videos from the aesthetic and technical perspectives, given its success. DOVER evaluates

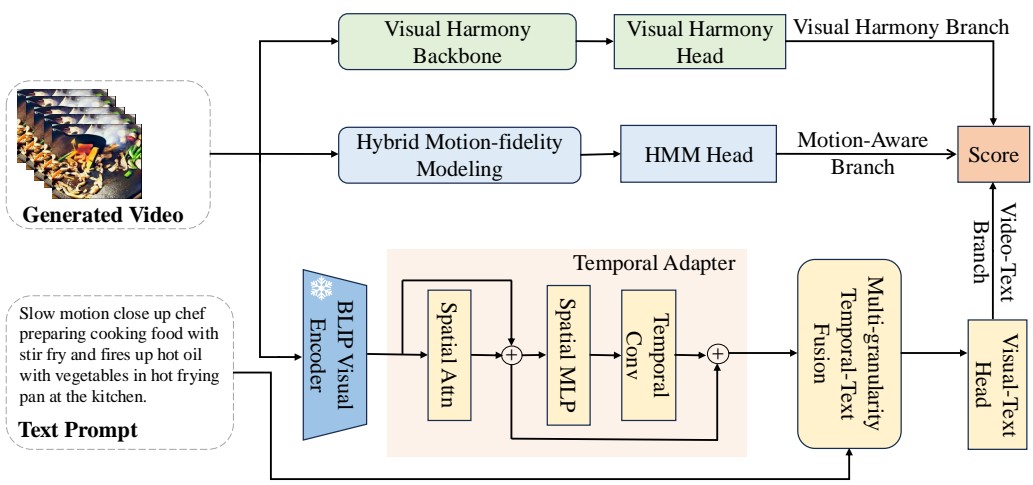

Figure 4: Network overview of the proposed CRAVE.

videos based on two dimensions: aesthetic score and technical distortion. In this work, we use the pre-trained DOVER method as the Visual Harmony Backbone, followed by a linear head to obtain the output, which could be formulated as:

$$F_{aes} = \Phi_{aes}(V), \tag{2}$$

$$F_{tech} = \Phi_{tech}(V), \tag{3}$$

$$O_{vh} = \omega_{vh}(F_{aes} \oplus F_{tech})), \tag{4}$$

where $V$ is the input video, $\Phi_{aes}$, $\Phi_{tech}$ represent the aesthetic encoder and distortion encoder in DOVER, respectively, $\oplus$ denotes concatenation along the dimension, $\omega_{vh}$ represents the linear head for this branch, and $O_{vh}$ refers to the corresponding output.

## 4.3 MULTI-GRANULARITY TEXT-TEMPORAL FUSION

Multi-granularity Text-Temporal (MTT) module firstly leverages high-quality priors from multi-modal understanding approaches like BLIP Li et al. (2022b), successfully extending it in the temporal dimension via effective temporal adapters. Then the visual information aggregated in the temporal adapter interacts with the text embedding via cross-attention. To flexibly fuse effective information from the text, we additionally perform multi-granularity aggregation on the text. Namely, in addition to the whole input, we break down the text into phrases and words of varying granularity containing different levels of semantic information via SpaCy Honnibal et al. (2020). After that, the integrated text embeddings of varying granularity are measured with the output of the visual branch, as shown in Figure 5. The entire process can be formulated as:

$$F_v = \Phi_t(\Phi_s(V)), \tag{5}$$

$$F_e = \Phi_e([enc; F_v], e), \tag{6}$$

$$F_{\text{enc}} = F_e[0, \ldots], \tag{7}$$

$$F_{\text{word}} = F_e[1 :, \ldots], \tag{8}$$

where $V$, $e$, $F_v$ are the input video, text prompt, and the derived visual feature, respectively. $\Phi_s$, $\Phi_t$, $\Phi_e$ denote the spatial encoder, temporal adapter, and text encoder, respectively. $F_{\text{enc}}$ and $F_{\text{word}}$ refer to the $[enc]$ token embedding and the word token embedding, representing different levels of semantics of the textual prompt. To get the phrase-level encoding for prompts, we utilize the successful word-to-phrase module in Zhu et al. (2023). The module converts the mapping between word and phrase to the mapping between word embedding and phrase embedding, based on the position mapping between word and the corresponding embedding, which could be formulated as,

$$\{p_1, p_2 \ldots, p_m\} = \Phi_w(\{w_1, w_2 \ldots, w_n\}), \tag{9}$$

$$\{F_{p_1}, F_{p_2} \ldots, F_{p_m}\} = \Phi_{F_w}(\{F_{w_1}, F_{w_2} \ldots, F_{w_n}\}), \tag{10}$$

where $\Phi_w$ and $\Phi_{F_w}$ refer to the mapping between word and phrase and the mapping between word embedding and phrase embedding. $p_i$, $w_i$ refers to the $i$-th useful phrase and word in prompt, while $F_{p_i}$, $F_{w_i}$ refers to the $i$-th phrase embedding and word embedding, respectively. After that, we calculate the cosine distance between all text features and video features, respectively. Finally, we sum them up to get the final score.

$$O_{align} = \sum_l cos(F_l, F_v), \tag{11}$$

where $l$ denotes levels of granularity such as the whole paragraph, phrase, and word level.

### 4.4 HYBRID MOTION-FIDELITY MODELING

Compared with natural videos, AIGC videos usually contain unique distortions such as irregular motions that violate the physical laws. Despite improvements in recent video generation models, low-fidelity motion remains a persistent challenge. Here, motions that defy logic, deformed motions, and motions with abnormal amplitudes are collectively referred to as "low-quality" motions. To better assess motion distortions in current AIGC videos, we propose Hybrid Motion-fidelity Modeling (HMM), which hierarchically captures motion features at different granularities. Specifically, given the success of optical flow in anomaly detection Caldelli et al. (2021); Agarwal et al. (2020), we leverage

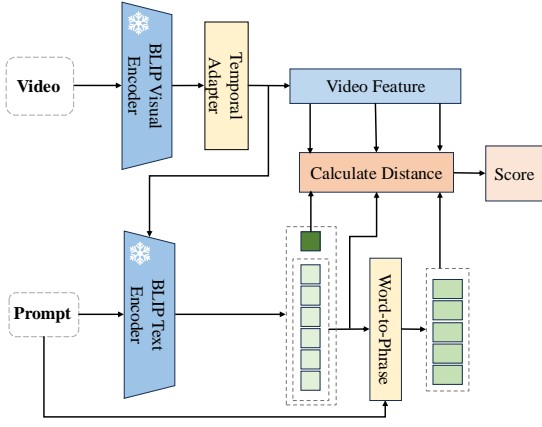

Figure 5: Details of the proposed MTT module.

dense motion information derived from flow to capture low-level motion patterns, combined with global abstract motion information from action recognition Kay et al. (2017); Goyal et al. (2017). Section 5 and Appendix E later demonstrate the effectiveness of combining these two aspects. In practice, flow features are extracted using the pre-trained StreamFlow Sun et al. (2024c), while high-level abstract motion priors are obtained from the pre-trained Uniformer Li et al. (2023). Different branches are then fused via a feed-forward network. A linear head is then employed to process the fused results and perform regression to generate the final output.

### 4.5 SUPERVISION

Based on the previous training objective in Wu et al. (2023a; 2022); Sun et al. (2022), the mix of rank loss Gao et al. (2019) and PLCC (Pearson Linear Correlation Coefficient) loss is adopted to train the overall network. Note that the BLIP visual and text encoder remains frozen throughout the training process. The whole training objective could be formulated as,

$$\mathcal{L} = \mathcal{L}_{plcc} + \gamma \cdot \mathcal{L}_{rank}, \tag{12}$$

where $\gamma$ is the coefficient set to 0.3 in the experiments.

## 5 EXPERIMENTS

### 5.1 EXPERIMENTAL SETUP

We use T2VQA-DB Kou et al. (2024b), GAIA Chen et al. (2024), VideoGenEval Zeng et al. (2024), and the proposed CRAVE-DB for evaluation. T2VQA-DB is the largest AIGC VQA dataset for text-driven video generation. It contains many AIGC videos from classic methods, which provides a good complement to the evaluation. We further perform cross-dataset tests on GAIA and VideoGenEval. As discussed in Section 3.2, since VideoGenEval does not have annotations, we use the Hunyuan and Wan subset of VideoGenEval as the test set and conduct subjective annotations following the ITU standard. During training and evaluation, we follow the same settings in DOVER Wu et al. (2023a).

### 5.2 QUANTITATIVE RESULTS

As shown in Table 2 and 3, we can learn that the leading performance of CRAVE on both the proposed content-rich dataset and the T2VQA-DB that includes prior AIGC videos. On CRAVE-DB, CRAVE demonstrates a particularly significant lead, highlighting its effectiveness in evaluating next-generation AIGC videos. On T2VQA-DB, CRAVE also outperforms previous models, even surpassing LLM-based models such as Q-Align and T2VQA, which further demonstrates the effectiveness of its design. "Ft." denotes methods that need further fine-tuning on the target dataset. In Table 3, "Bg.", "Sub.", "Consis", "Aes.", "Sm." denotes

Table 2: Quantitative comparison on T2VQA-DB.

| Type | Models | SRCC | PLCC | KRCC |
|---|---|---|---|---|
| 0-shot | CLIPSim Radford et al. (2021) | 0.1047 | 0.1277 | 0.0702 |
| | BLIP Li et al. (2022b) | 0.1659 | 0.1860 | 0.1112 |
| | ImageReward Xu et al. (2023) | 0.1875 | 0.2121 | 0.1266 |
| | ViCLIP Wang et al. (2023e) | 0.1162 | 0.1449 | 0.0781 |
| | UMTScore Liu et al. (2023) | 0.0676 | 0.0721 | 0.0453 |
| Ft. | SimpleVQA Sun et al. (2022) | 0.6275 | 0.6338 | 0.4466 |
| | BVQA Li et al. (2022a) | 0.7390 | 0.7486 | 0.5487 |
| | FastVQA Wu et al. (2022) | 0.7173 | 0.7295 | 0.5303 |
| | DOVER Wu et al. (2023a) | 0.7609 | 0.7693 | 0.5704 |
| | Q-Align Wu et al. (2023b) | 0.7601 | 0.7768 | 0.5860 |
| | T2VQA Kou et al. (2024b) | 0.7965 | 0.8066 | 0.6058 |
| Ours | CRAVE | **0.8122** | **0.8214** | **0.6338** |

the background, subject, consistency, aesthetic, and smoothness, respectively. It can be seen that 0-shot methods tend to have lower results, which is also observed in previous works Kou et al. (2024b); Sun et al. (2024b). It could be due to the lack of alignment with human perception or the consideration of the dynamic distortion in AIGC videos. We further include comparisons in GPU memory and latency in Appendix D, which demonstrate the efficiency of CRAVE.

## 5.3 QUALITATIVE RESULTS

We first visualize the difference between the predicted and the ground-truth MOS, as illustrated in Appendix F. The curves are obtained from a fourth-order polynomial nonlinear fitting. We then further present cases of CRAVE-DB and limitations of CRAVE in Appendix F and G, which include the scores of various samples by different quality assessment models and failure cases of CRAVE, respectively. It can be seen that models trained on CRAVE-DB demonstrate more accurate evaluations for those cases compared to training on T2VQA-DB, indicating that CRAVE-DB serves as an effective complement to existing datasets. More video cases could be found in the **supplemental materials**.

Table 3: Quantitative comparison on CRAVE-DB.

| Type | Models | SRCC | PLCC | KRCC |
|---|---|---|---|---|
| 0-shot | UMTScore Liu et al. (2023) | 0.0134 | 0.0355 | 0.0118 |
| | Flicker Huang et al. (2024b) | 0.0761 | 0.0687 | 0.0493 |
| | Bg. Consis. Huang et al. (2024b) | 0.1170 | 0.0829 | 0.0773 |
| | Sub. Consis. Huang et al. (2024b) | 0.1418 | 0.0956 | 0.0936 |
| | ViCLIP Wang et al. (2023e) | 0.1290 | 0.1207 | 0.0857 |
| | LongCLIP Zhang et al. (2024) | 0.2757 | 0.3033 | 0.1884 |
| | Aes. Score Huang et al. (2024b) | 0.3557 | 0.3457 | 0.2373 |
| | HPSv2 Wu et al. (2023c) | 0.0562 | 0.0501 | 0.0391 |
| | PickScore Kirstain et al. (2023) | 0.0557 | 0.0492 | 0.0361 |
| | CHScore Yuan et al. (2024) | 0.2202 | 0.1628 | 0.1470 |
| | ImageReward Xu et al. (2023) | 0.2216 | 0.2125 | 0.1470 |
| | Motion Sm. Huang et al. (2024b) | 0.2331 | 0.2630 | 0.1549 |
| Ft. | SimpleVQA Sun et al. (2022) | 0.6230 | 0.6180 | 0.4436 |
| | StableVQA Kou et al. (2023) | 0.6396 | 0.6415 | 0.4552 |
| | Q-Align Wu et al. (2023b) | 0.6481 | 0.6492 | 0.4712 |
| | FastVQA Wu et al. (2022) | 0.7155 | 0.7062 | 0.5243 |
| | DOVER Wu et al. (2023a) | 0.7095 | 0.7192 | 0.5224 |
| | TriVQA Qu et al. (2024) | 0.7183 | 0.7313 | 0.5293 |
| | T2VQA Kou et al. (2024b) | 0.7266 | 0.7098 | 0.5369 |
| Ours | CRAVE | **0.7587** | **0.7581** | **0.5660** |

## 5.4 CROSS-DATASET VALIDATION

In this section, all models are trained on T2VQA-DB given its large capacity. They are then performed 0-shot testing on CRAVE-DB, GAIA, and VideoGenEval, as shown in Table 4. We choose GAIA as it is one of a limited number of public ITU-Compliant AIGC VQA datasets (Annotators no less than 15). We chose VideoGenEval based on the discussion in Section 3.2. Since VideoGenEval does not have labels, we conduct annotations based on settings in Section 3.3. Considering the annotation cost and the distinctiveness from other datasets, we selected videos from the latest video generation models Wan and Hunyuan subsets, which include 465 samples. It can be seen that CRAVE demonstrates excellent cross-dataset generalization capability across different test sets.

## 5.5 ABLATION STUDY

We further ablate each component of CRAVE, as shown in Table 5. Underlined settings are used in our final model. As CRAVE-DB naturally contains intricate texts, rich motion information, and other

Table 4: Cross-dataset validation on CRAVE-DB, GAIA, and VideoGenEval.

| Method | CRAVE-DB | | GAIA | | VGenEval-Hunyuan | | VGenEval-Wan | |
|---|---|---|---|---|---|---|---|---|
| | SRCC | PLCC | SRCC | PLCC | SRCC | PLCC | SRCC | PLCC |
| DOVER Wu et al. (2023a) | 0.3480 | 0.3719 | 0.1187 | 0.1141 | 0.4363 | 0.3663 | 0.3284 | 0.3259 |
| Q-Align Wu et al. (2023b) | 0.3757 | 0.3943 | 0.2105 | 0.1747 | 0.4331 | 0.4335 | 0.2943 | 0.3013 |
| T2VQA Kou et al. (2024b) | 0.4064 | 0.4166 | 0.2198 | 0.2166 | 0.4340 | 0.4498 | 0.3570 | 0.3785 |
| CRAVE (Ours) | **0.4213** | **0.4326** | **0.2263** | **0.2258** | **0.4595** | **0.4694** | **0.3853** | **0.3805** |

Table 5: Quantitative results of ablation study.

| Experiment | Method | CRAVE-DB | | | T2VQA-DB | | |
|---|---|---|---|---|---|---|---|
| | | SRCC | PLCC | KRCC | SRCC | PLCC | KRCC |
| Temporal-Text Fusion | None | 0.6512 | 0.6601 | 0.4643 | 0.7701 | 0.7804 | 0.5885 |
| | ST-Graph. | 0.6966 | 0.6968 | 0.5042 | 0.7990 | 0.8084 | 0.6196 |
| | Temp. Attn. | 0.7159 | 0.7175 | 0.5226 | 0.7989 | 0.8089 | 0.6199 |
| | Pseudo 3D Conv | **0.7310** | **0.7241** | **0.5335** | **0.8077** | **0.8167** | **0.6289** |
| Multi-Granularity Text Injection | None | 0.7477 | 0.7458 | 0.5526 | 0.8077 | 0.8167 | 0.6289 |
| | Global | 0.7520 | 0.7485 | 0.5566 | 0.8115 | 0.8207 | 0.6329 |
| | Local | 0.7524 | 0.7498 | 0.5578 | 0.8121 | 0.8212 | 0.6337 |
| | Individual | 0.7491 | 0.7460 | 0.5534 | 0.8111 | 0.8200 | 0.6326 |
| | Combined | **0.7587** | **0.7581** | **0.5660** | **0.8122** | **0.8214** | **0.6338** |
| Motion-Aware Modeling | None | 0.7507 | 0.7499 | 0.5561 | 0.8103 | 0.8199 | 0.6317 |
| | High-Level | 0.7527 | 0.7529 | 0.5605 | 0.8111 | 0.8204 | 0.6326 |
| | Low-Level | 0.7529 | 0.7513 | 0.5576 | 0.8118 | 0.8211 | 0.6330 |
| | Hybrid | **0.7587** | **0.7581** | **0.5660** | **0.8122** | **0.8214** | **0.6338** |
| Flow Frames | 4 | 0.7562 | 0.7552 | 0.5598 | 0.8110 | 0.8206 | 0.6323 |
| | 8 | 0.7566 | 0.7564 | 0.5612 | 0.8113 | 0.8208 | 0.6326 |
| | 16 | **0.7587** | **0.7581** | **0.5660** | **0.8122** | **0.8214** | **0.6338** |

such content, we could learn that the improvements on it are generally more significant. We first explore ways to align the text with temporal visual features. ST-Graph denotes the spatio-temporal graph modeling, which flattens the temporal dimension into the spatial. Temp. Attn. is the attention operator along the additional temporal dimension. Pseudo 3D Conv Singer et al. (2023) stacks additional convolutions in the temporal dimension. We can see that the effectiveness has significantly improved with temporal modeling, and that the Pseudo 3D Conv widely used in generation tasks also excels in long-text spatiotemporal modeling. We then investigate the granularity of MTT and discover that integrating all granularity levels yields optimal performance. A more detailed analysis for each branch is in Appendix E. Additionally, we examine the impact of motion-aware temporal modeling. Our experiments demonstrate that dense data from optical flow enhances overall performance, and incorporating sparse abstract spatiotemporal information provides a significant performance boost. We further explored the impact of flow frames. We observed that using more optical flow frames tends to improve accuracy. Given the trade-off between accuracy and efficiency, we ultimately chose 16 frames for the flow calculation. Besides, we ablate the effect of various backbones in visual harmony, visual-text alignment, and motion-aware modeling. Please refer to Appendix E for details.

## 6 CONCLUSION

Given the gap between concurrent text-driven video generation models and the existing AIGC VQA dataset, we introduce CRAVE, an effective VQA method, and CRAVE-DB, a new benchmark for the next-generation AIGC videos. Based on the effective multi-dimensional design, CRAVE achieves excellent human-aligned results across multiple metrics and datasets. CRAVE-DB better aligns with contemporary AIGC-generated videos, serving as an effective complement to existing datasets.

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

## A    ETHICS STATEMENT

This work is conducted in strict adherence to the ICLR Code of Ethics. We affirm our commitment to contributing to societal and human well-being by focusing our research on a socially beneficial task. The utilized datasets and models involves no human subjects directly, and thus carries no risks to personal privacy, health, or safety. There is no discrimination or negative societal impact in the subjective or objective experimental phases, and all findings will be accurately reported.

## B    REPRODICIBILITY STATEMENT

To ensure reproducibility, this work provides detailed descriptions of the experimental procedures. The proposed dataset, code, and model weights used will be made publicly available. In Section 3 and Appendix H, we elaborate on the dataset construction and the details of the subjective experiments, while Section 5 specifies the model parameters. The T2VQA-DB dataset utilized is publicly accessible. We believe this information will facilitate the reproduction of our work and help advance the field.

## C    USE OF LLMS

Large Language Models (LLMs) were used in this work solely for typo checking and language polishing. It is important to note that LLMs were not involved in the ideation, research methodology, or experiment design of this work. All research concepts, ideas, experiments, and analyses were conducted by the authors. The authors take full responsibility for the content of the manuscript, including the text polished by LLMs. We have verified that the LLM-generated text adheres to ethical guidelines and is free from issues such as plagiarism or scientific misconduct.

## D    COMPARISON ON MODEL EFFICIENCY

In this section, we compare the computational costs (inference speed, GPU memory usage, and parameter count) of different quality assessment models, as shown in Table 6. The original video input resolution is 1280×720, and the tests were conducted on NVIDIA RTX 3090 GPUs. The reported time is the average result of 5 measurements. From the table, it could be seen that CRAVE, achieves superior alignment performance compared to methods with equal or even higher parameter counts and more powerful multimodal foundation models. Furthermore, CRAVE does not fall behind in terms of inference speed, memory usage, and parameter efficiency.

Table 6: Comparison on inference speed, GPU memory usage, and parameter count.

| Method | GPU Memory | Parameter Counts | FPS | SRCC | PLCC |
|---|---|---|---|---|---|
| Dover | **3.5G** | **58.1M** | **63.5** | 0.7095 | 0.7192 |
| StableVQA | 6.1G | 118.0 M | 61.8 | 0.6396 | 0.6415 |
| TriVQA | 7.7G | 253.8M | 56.3 | 0.7183 | 0.7103 |
| Q-Align | 21.5G | 5B | 21.8 | 0.6481 | 0.6492 |
| T2VQA | 16.2G | 7B | 36.4 | 0.7266 | 0.7098 |
| CRAVE (Ours) | 10.8G | 731.7M | 50.1 | **0.7587** | **0.7581** |

## E    ADDITIONAL ABLATED RESULTS

In this section, we first further explore the impact of each branch in MTT, as shown in Table 7. We observe that varying semantic granularity brought by a single branch can influence the vision-language alignment differently. For example, as shown in the first two rows, replacing Local + Individual branches with Local + Local, the performance will drop due to the absence of individual-level granularity. If we further replace the Global branch that provides global information, with the

Individual branch, and downgrade Local information to Individual information, the performance further drops, even though these experiments have the same number of branches and parameters. This highlights the necessity of multi-granularity.

Table 7: Ablation study on MTT branches.

| Method | SRCC | PLCC |
|---|---|---|
| Glboal + Location + Individual | 0.7587 | 0.7581 |
| Glboal + Location + Location | 0.7554 | 0.7567 |
| Individual + Location + Location | 0.7552 | 0.7527 |
| Individual + Individual + Individual | 0.7435 | 0.7473 |

To better illustrate how hybrid motion modeling works, we further divide the CRAVE dataset. Following EvalCrafter Liu et al. (2024), we use Flow-Score (average optical flow magnitude of videos) to measure video dynamics and split the test set into low-dynamic (Flow-Score less than 5) and high-dynamic (Flow-Score ¿ 5) subsets. The performance of different modeling methods is presented in Table 8. We find that CRAVE shows excellent performance even in highly dynamic regions. Meanwhile, low-level encoders like optical flow are more sensitive to higher dynamic videos and thus perform better on the high set. In comparison, high-level action recognition backbones might emphasize action semantics, resulting in relatively less improvement in higher dynamics compared to the low-level encoder. It could also be seen that a hybrid approach combining both methods effectively leverages their complementary strengths.

Table 8: Ablation study on subset of CRAVE-DB.

| Method | Low Set | | High Set | |
|---|---|---|---|---|
| | SRCC | PLCC | SRCC | PLCC |
| High-Level only | 0.7301 | 0.7232 | 0.7589 | 0.7650 |
| Low-Level only | 0.7241 | 0.7197 | 0.7914 | 0.7941 |
| Hybrid | **0.7322** | **0.7231** | **0.7919** | **0.8011** |

We then further verify the choice of different backbones in visual harmony, visual-text alignment, high-level and low-level motion modeling, as shown in Table 9. It could be seen that the quality of the pre-trained model influences the final results within a certain range, but does not cause particularly significant fluctuations in performance.

Table 9: Ablation study on various backbones.

| Experiment | Method | SRCC | PLCC |
|---|---|---|---|
| Visual Harmony Backbone | SimpleVQA Sun et al. (2022) | 0.7519 | 0.7485 |
| | TriVQA Qu et al. (2024) | 0.7547 | 0.7556 |
| | DOVER Wu et al. (2023a) | **0.7587** | **0.7581** |
| Visual-Text Alignment | CLIP Radford et al. (2021) | 0.7460 | 0.7302 |
| | DIFT Tang et al. (2023) | 0.7347 | 0.7365 |
| | BLIP Li et al. (2022b) | **0.7497** | **0.7501** |
| High-Level Motion Modeling | MVD Wang et al. (2023b) | 0.7470 | 0.7536 |
| | Uniformer Li et al. (2023) | **0.7587** | **0.7581** |
| Low-level Motion Modeling | Sea-RAFT Wang et al. (2024) | 0.7540 | 0.7527 |
| | StreamFlow Sun et al. (2024c) | **0.7587** | **0.7581** |

We also ablate the effect of the initial gamma values introduced in Section 4.5, as shown in Table 10. The initial gamma value was set following DOVER Wu et al. (2023a). We tested other values to further validate our choice. It can be seen that gamma moderately affects performance.

Table 10: Ablation study on gamma values.

| Gamma | SRCC | PLCC |
|-------|------|------|
| 0.1 | 0.7583 | 0.7506 |
| 0.3 | **0.7587** | **0.7581** |
| 0.5 | 0.7507 | 0.7479 |

## F  ADDITIONAL QUALITATIVE RESULTS

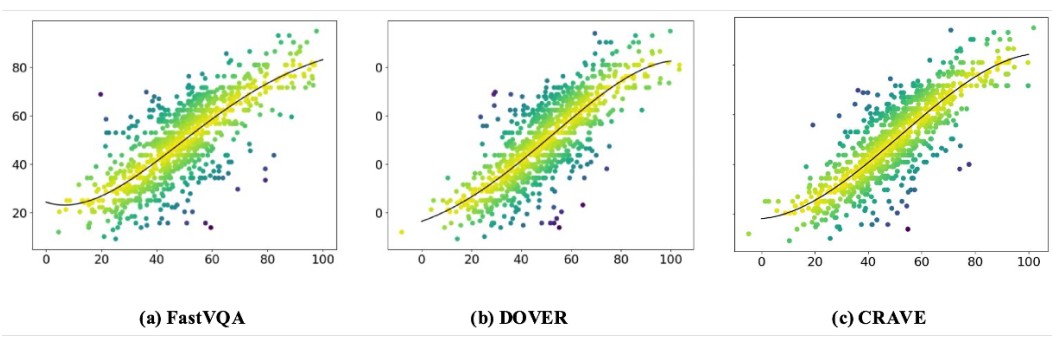

(a) FastVQA          (b) DOVER          (c) CRAVE

Figure 6: Scatter plots of the predicted scores and ground-truth MOSs. A brighter scatter point represents higher density.

In this section, we first visualize the difference between predicted and ground-truth MOS. As shown in Figure 6, (a), (b), and (c) represent the visualization of the differences between different models on T2VQA-DB. The more clustered the points, the smaller the differences. We can observe that the points in (a) and (b) are more scattered and farther from the central line. A fourth-order polynomial nonlinear fitting is used to draw the central line.

We further present the challenges of existing methods on the constructed dataset to show the necessity of CRAVE-DB. As shown in Figure 7, it provides detailed failure cases of multiple previous methods on the new dataset. We found that these cases often require more complex semantic alignment or occur under more dynamic conditions. Even state-of-the-art methods like T2VQA and classic approaches like Dover tend to make mistakes, suggesting that following methods could further improve consistency in complex text and motion alignment. However, after training on CRAVE-DB, **these biases are significantly reduced, which demonstrates CRAVE-DB effectively complements T2VQA-DB**.

## G  LIMITATIONS AND FAILURE CASES

Overall, CRAVE also has its own limitations. First, despite CRAVE's multiple rounds of manual screening and the use of an above-average number of annotators to ensure the diversity and reliability of the data, its applicability still faces challenges when compared to complex real-world scenarios. Second, given the current network design paradigm, the number of frames and resolution used by CRAVE during training are still limited, which may still create a gap when compared to actual videos. We believe that the evaluation of text-driven video generation tasks still has many pressing issues that need to be resolved. Although CRAVE has made some improvements compared to previous methods, there are still many problems waiting to be addressed in follow-up work. As shown in Figure 8, CRAVE still has room for improvement in evaluating long textual descriptions and local distortions. For some physical activities, the assessments made by CRAVE may still have biases in certain situations.

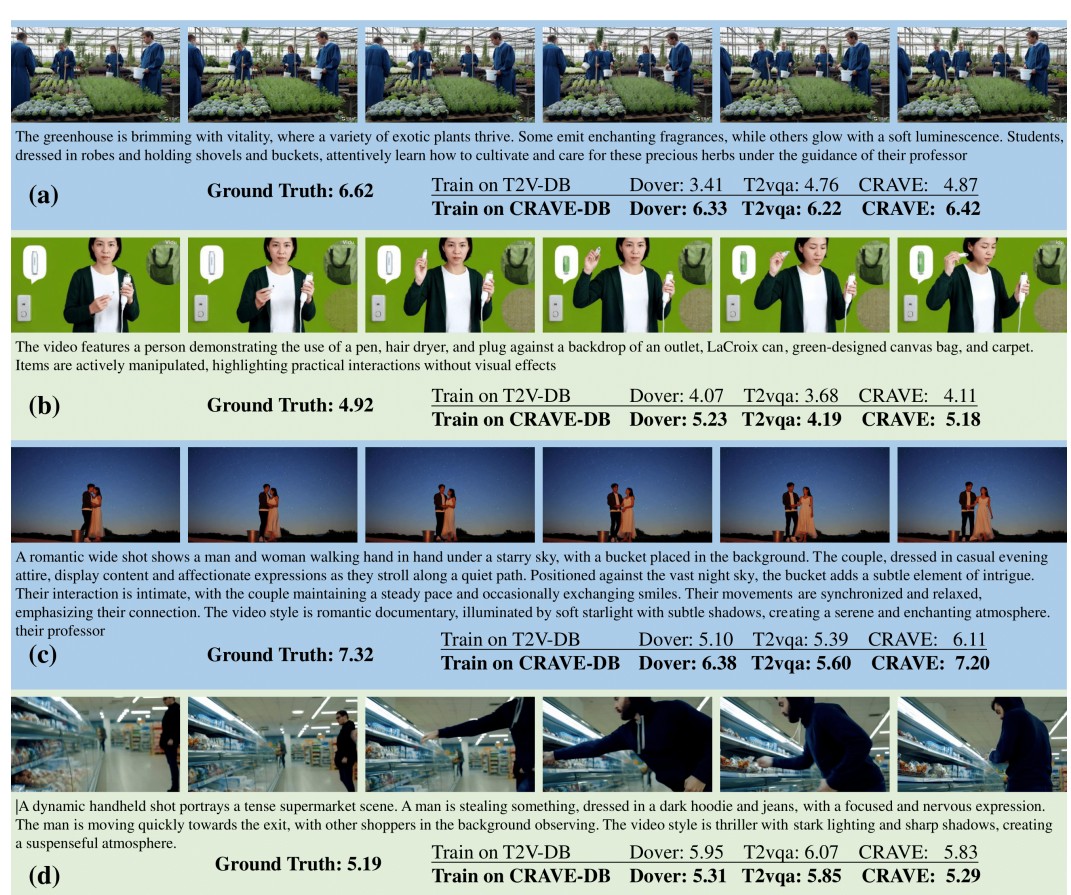

Figure 7: Samples and corresponding scores from models trained on different datasets.

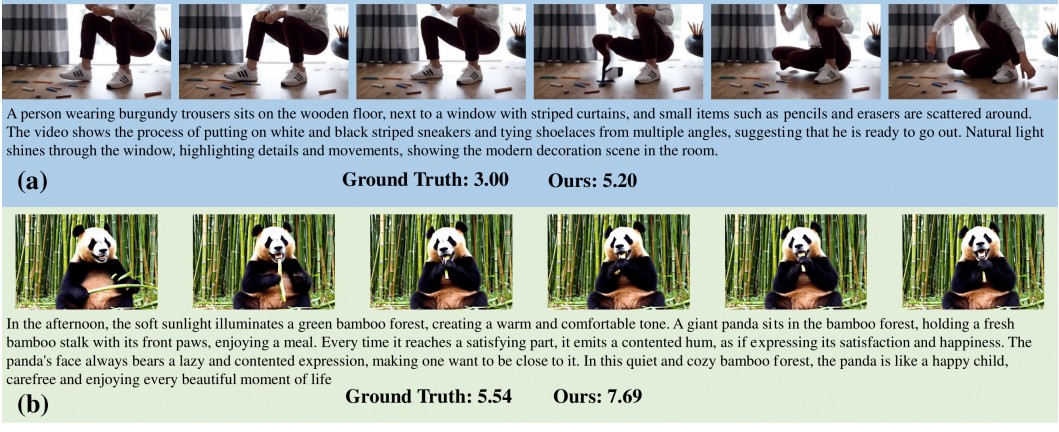

Figure 8: Failure cases of CRAVE.

# H    DETAILS OF SUBJECTIVE STUDY

In this section, we give a more detailed discussion of the subjective experiments. The subjective experiments were conducted in two batches. The software interface used in the subjective experiments is illustrated in Figure 9. During the scoring, the annotators were advised to assess the video from 3 perspectives: (1) visual quality, which is commonly used in traditional VQA methods; (2) matching between the text and video; and (3) motion quality, such as whether motion consistency is maintained, whether the motion is distorted, and whether it aligns with common sense. The first batch involved annotating the CRAVE-DB dataset, with 29 participants. In this batch, all videos were divided into 10 sessions. The second batch focused on annotating the test set of VideoGenEval, involving 31 participants. All videos in this batch were divided into 5 sessions. On average, each participant spent approximately 0.91 hours completing one session. Each participant was compensated $8.2 per session in accordance with current ethical standards Silberman et al. (2018). The experiments contained no NSFW (Not Safe For Work) or other inappropriate content in the text prompts or generated outputs. Participants came from diverse academic backgrounds and were trained using out-of-dataset cases prior to formal scoring to ensure annotation consistency.

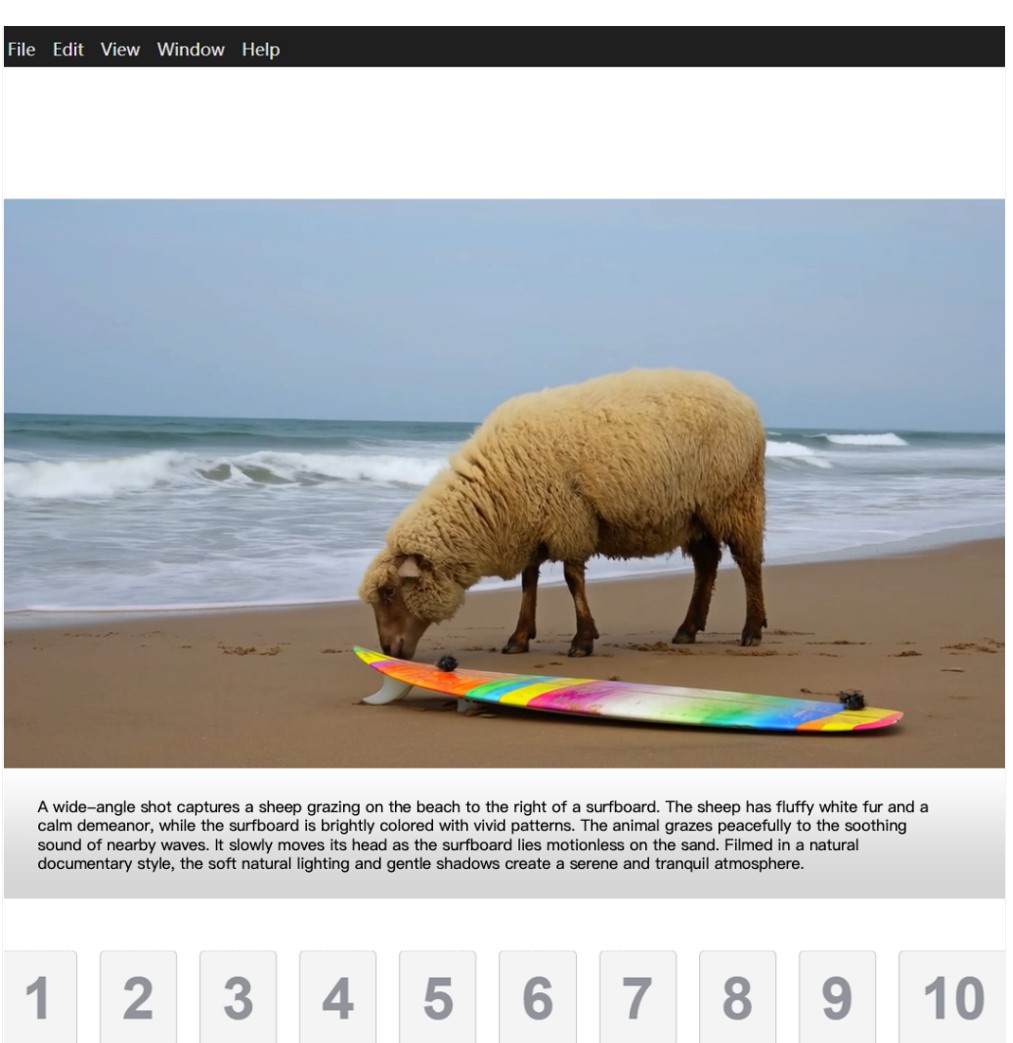

Figure 9: User Interface for the subjective experiment.

