# OpenReview forum: "Content-Rich AIGC Video Quality Assessment via Intricate Text Alignment and Motion-Aware Consistency"
_ICLR.cc/2026/Conference — Submitted to ICLR 2026_

### Official Review · Reviewer_Dwqz · 2025-10-19

**Soundness:** 2
**Presentation:** 2
**Contribution:** 2
**Rating:** 4
**Confidence:** 5

**Summary:**

This paper proposes CRAVE for content-rich AIGV evaluation, which adopts multi-granularity text-temporal fusion and leverages the hybrid motion-fidelity modeling to assess temporal artifacts. Meanwhile, a dataset named CRAVE-DB is proposed to evaluate AIGV generated using elaborate prompts. Experiments show that CRAVE surpasses baseline models on several AIGV datasets, achieving more human-aligned predictions.

**Strengths:**

1. The research motivation is clear. For videos generated based on complex content prompts, an accurate evaluator is needed.
2. An newer AIGV dataset generated with content-rich prompts and a corresponding video evaluator.

**Weaknesses:**

1. Although the proposed CRAVE-DB contains prompts with more words, its size is limited compared to existing AIGV datasets, with only 410 prompts and 1,228 videos generated from only 4 models.
2. The contributions to the dataset and the method are both limited, which mainly involves the integration of existing methods.
3. More recent AIGV benchmark should be compared (2024-2025), considering the timeliness of this direction.
4. Lack of detailed data analysis of the subjective experimental results, such as the outliers, inter-user correlations, cross-model MOS performance, etc.
5. The expression of the article needs to be improved. Some descriptions are overly repetitive, such as the ITU standards which is merely a necessary factor to consider, but it is not the main point.
6. The model combines three aspects into a single score to describe AIGV. Does this violate the aforementioned fine-grained characterization? Are these three dimensions highly correlated or independent? Further analysis is needed on this.

**Questions:**

1. The presentation of Fig. 1 is somewhat vague, where I can only observe that the AIGC videos of the new generation contain more comprehensive content, which indicates that the prompt descriptions are more detailed. So, how would Lavie perform if the same prompt was used? The comparison should be made under the same conditions.
2. Could the describtion of “the gap between the naturalness and complexity of the latest AIGC videos and previous videos has become markedly apparent” be more detailed? or provide analysis support?
3. Tab. 1 could include more information, such as #video, # prompts, # models, as provided in Section 2.3, to enhance readability.
4. The description in line 161 “Each video has a duration of over 5 seconds with a fps of 24” may not comprehensive enough, as stated in Line 123-124 that current models generate videos with a duration of more than 5 seconds and frame rates above 24 fps. More long videos with high frame rate is worth further investigation.
5. Improper use of double quotes in line 194: ”landscape”, line 195: ”animal”, and line 196 ”object”.
6. It would be more helpful to provide an intuitive example of a generated fine-grained prompt in Figure 2.
7. The considered three evaluation perspectives: visual quality, T2V alignment, and motion quality are common and fair.
8. I suggest that the three dimensions that need to be evaluated be presented in the main text. This is quite important, especially for explaining the MOS distribution shown in Figure 3. Currently, I'm not sure what MOS in Figure 3 represents.
9. The description in Figure 4 could be more straightforward. Presenting the network structure directly rather than using a vague term would be more helpful for readers to understand, especially for the relatively simple modules.
10. It is suggested to conduct a detailed visualization of SpaCy's and the MTT’s working principle or integrate it into the existing Figure 4 (Line 308).
11. The operation mechanism of the HYBRID MOTION-FIDELITY MODELING module also requires a detailed illustration rather than a simple flowchart in current Fig. 4.
12. Lack of statistical tests for the significance of performance differences, as it seem that the prediction results on DOVER are very similar to those on CRAVE (Fig. 6).

---

### Official Review · Reviewer_KLcS · 2025-10-20

**Soundness:** 2
**Presentation:** 2
**Contribution:** 2
**Rating:** 4
**Confidence:** 5

**Summary:**

This paper addresses the problem of evaluating next-generation text-to-video (T2V) content produced by Sora-era AIGC models. It introduces CRAVE, a model for content-rich AIGC video quality assessment that combines multi-granularity text-temporal (MTT) fusion and hybrid motion-fidelity modeling (HMM). It introduces CRAVE-DB, a new ITU-compliant benchmark of long, detailed prompts and videos from models such as Sora, Kling, and Vidu.
Experiments across T2VQA-DB, GAIA, VideoGenEval, and the new CRAVE-DB show state-of-the-art correlations with human ratings.

**Strengths:**

The paper tackles a timely and relevant problem by addressing the quality assessment of next-generation AIGC videos that contain richer semantics, longer durations, and more complex motion dynamics. It makes a meaningful empirical contribution by constructing CRAVE-DB, one of the first ITU-compliant datasets that includes content from cutting-edge video generators such as Sora, Kling, and Vidu. The proposed CRAVE framework achieves good results in the experiment.

**Weaknesses:**

1) The proposed CRAVE is not that new. The BLIP backbone, temporal adapter, and optical-flow features are well-known. The combination is logical but may be seen as incremental engineering rather than conceptual innovation. The novelty primarily lies in benchmark construction rather than architecture. It is suggested to better clarify the novelty of the proposed method.

2) Although CRAVE-DB is ITU-compliant, it contains only ≈ 1.2 K videos, smaller than existing general-purpose sets like T2VQA-DB (10 K videos). Claims of “large-scale benchmark” may be overstated. Furthermore, the number of video generators is limited, which raises concerns about the diversity of the proposed dataset. More comparisons should be made on large-scale datasets from VideoScore: Building Automatic Metrics to Simulate Fine-grained Human Feedback for Video Generation, Q-Eval-100K: Evaluating Visual Quality and Alignment Level for Text-to-Vision Content.

**Questions:**

Please see the weaknesses.

---

### Official Review · Reviewer_pU76 · 2025-11-01

**Soundness:** 3
**Presentation:** 3
**Contribution:** 3
**Rating:** 6
**Confidence:** 4

**Summary:**

This paper introduces CRAVE, a purpose-built quality evaluator for the emerging generation of text-to-AIGC videos that exhibit longer duration, richer semantics, and more complex motion than their predecessors. By integrating (i) a visual-harmony branch that borrows DOVER’s aesthetic and distortion priors, (ii) a Multi-granularity Text-Temporal fusion module that aligns long, elaborate prompts with video dynamics via word-, phrase-, and sentence-level cross-attention, and (iii) a Hybrid Motion-fidelity Modeling component that jointly exploits dense optical-flow and high-level action semantics, CRAVE produces a single human-aligned quality score. Coupled with CRAVE-DB—a new 1 228-video benchmark annotated by 29 subjects according to ITU standards—the framework achieves state-of-the-art correlation with human ratings on both legacy (T2VQA-DB) and next-generation datasets, and generalises zero-shot to unseen models.

**Strengths:**

1. A three-branch architecture that explicitly disentangles and fuses visual harmony, fine-grained text–video alignment, and hierarchical motion fidelity for content-rich AIGC-VQA.
2. CRAVE-DB, the first publicly available benchmark populated exclusively by Sora-era models, offering long, detail-dense prompts and statistically reliable MOS labels.
3. Extensive experiments demonstrating superior SRCC/PLCC over existing metrics and a consistent zero-shot ranking of the latest generative models, validating CRAVE’s utility as a community standard.

**Weaknesses:**

1. Limited scale: The entire CRAVE-DB provides only 1 228 clips with ≈ 35 k human ratings—orders of magnitude smaller than the million-level weak-label sets commonly exploited in natural-video VQA—while the paper omits cross-dataset generalisation curves, leaving the community uncertain about over-fitting risks.

2. Prohibitive optical-flow cost: HMM computes dense optical flow across 16 frames, yielding 4–6× slower inference than single-frame baselines; neither FLOPs nor wall-clock latency is reported, and scalability to videos longer than 10 s remains unverified.

**Questions:**

1. The paper reports neither FLOPs, memory footprint, nor wall-clock latency for the dense 16-frame optical-flow branch; without such complexity metrics it is unclear whether CRAVE's gains are achievable in real-time or industrial pipelines, nor how accuracy degrades under resource-constrained scenarios. It's recommended to add.

2. Since competing baselines include image-oriented models (e.g., ImageReward, PickScore, etc.), a necessary ablation is to strip temporal components and evaluate CRAVE on established AIGC-IQA datasets (e.g., AIGIQA-20k); if the redesigned multi-granularity text–visual fusion still surpasses these image specialists, one can attribute the superiority to methodological innovation rather than to the rivals' inability to handle video inputs.

---

### Meta-Review · Area_Chair_rRJd · 2026-01-06

**Summary:**

The paper proposes CRAVE, a video quality assessment model designed for "content-rich" AI-generated videos (specifically targeting next-generation models like Sora), and introduces a corresponding benchmark, CRAVE-DB. The method utilizes a multi-granularity text-temporal fusion module and a hybrid motion-fidelity modeling approach to align complex prompts with video dynamics. While the reviewers acknowledged the timeliness of the problem and the effort to construct a new benchmark, the consensus is that the submission falls below the acceptance threshold. The primary concerns raised by the reviewers include limited novelty, dataset scale, and missing analysis. The authors did not provide a rebuttal in the discussion period. Hence, the paper was not recommended for acceptance.

**Reviewer Concerns:**

Addressed Concerns: None. The authors did not submit a rebuttal or participate in the discussion phase; therefore, none of the questions or concerns raised by the reviewers were addressed.

Outstanding Concerns: Since no feedback was provided by the authors, all issues raised in the reviews remain outstanding.

**Reviewer Scores:**

Reviewer pU76: 6 -> 4 (Downgrade)
Reviewer KLcS: 4 ->  4 (Unchanged)
Reviewer Dwqz: 4 ->  4 (Unchanged)

---

### Decision · Program_Chairs · 2026-01-26

Reject